# The Use of Glucagon-like Peptide-1 Receptor Agonists in Patients with Type 2 Diabetes Mellitus Does Not Increase the Risk of Pancreatic Cancer: A U.S.-Based Cohort Study

**DOI:** 10.3390/cancers16091625

**Published:** 2024-04-23

**Authors:** Mark Ayoub, Carol Faris, Tajana Juranovic, Harleen Chela, Ebubekir Daglilar

**Affiliations:** 1Department of Internal Medicine, Charleston Area Medical Center, West Virginia University, Charleston, WV 25304, USA; tajana.juranovic@camc.org; 2Department of General Surgery, Marshall University, Huntington, WV 25755, USA; farisc@marshall.edu; 3Division of Gastroenterology and Hepatology, Charleston Area Medical Center, West Virginia University, Charleston, WV 25304, USA

**Keywords:** diabetes, pancreatic cancer, GLP-1 receptor agonists

## Abstract

**Simple Summary:**

This is a retrospective national cohort study that aims to assess the risk of developing pancreatic cancer in patients with type 2 diabetes mellitus (T2DM) who are being treated with Glucagon-like Peptide-1 receptor agonists. We exclude patients with multiple risk factors for pancreatic cancer from our study. We retrospectively follow the patients for 7 years after the initiation of treatment. Current evidence is controversial about their use and the possible risk of pancreatic disease. We aim to revoke the association of pancreatic cancer with their use in support of their continued use due to their beneficial cardiovascular and renal effects.

**Abstract:**

Background: GLP-1 RAs are widely used for T2DM treatment due to their cardiorenal and metabolic benefits. This study examines the risk of pancreatic cancer with GLP-1 RA use in patients with T2DM. Methods: We analyzed TriNetX’s deidentified research database using the U.S. Collaborative Network comprising 62 healthcare organizations across the U.S.A. Patients with T2DM were split into two cohorts: one receiving GLP-1 RAs, and one not receiving GLP-1 RAs. We excluded patients with known risk factors for pancreatic cancer, including pancreatic cysts, a personal or family history of BRCA1, BRCA2, CDKN2A, KRAS, MEN1, MLH1, MSH2, NOTCH1, PALB2, PMS2, and PRSS1S genes, family history of pancreatic cancer, and VHL syndrome. Using a 1:1 propensity score-matching model based on baseline characteristics and comorbidities, we created comparable cohorts. We then compared the rate of pancreatic cancer between the two cohorts at a 7-year interval. Results: Out of 7,146,015 identified patients with T2DM, 10.3% were on a GLP-1 RA and 89.7% were not. Post-PSM, 721,110 patients were in each group. Patients on GLP-1 RAs had a 0.1% risk compared to a 0.2% risk of pancreatic cancer in the 7-year timeframe. Conclusion: The use of GLP-1 RAs in patients with type 2 diabetes mellitus (T2DM) does not appear to substantially elevate the risk of pancreatic cancer; in fact, it may potentially exert a protective effect.

## 1. Introduction

Pancreatic cancer poses a significant health concern globally due to its aggressive nature and poor prognosis. Notably, patients with type 2 diabetes mellitus (T2DM) have garnered attention as a potentially high-risk group for developing pancreatic cancer. Glucagon-like Peptide-1 receptor agonists (GLP-1 RAs) have emerged as a cornerstone in the management of type 2 diabetes mellitus (T2DM), offering glycemic control with favorable cardiovascular outcomes. However, recent observations and clinical studies have raised concerns regarding the potential association between GLP-1 RA use and the risk of pancreatic cancer [1,2]. This association has prompted extensive investigation into the safety profile of these medications, aiming to elucidate the underlying mechanisms and assess the true magnitude of risk. Understanding the implications of GLP-1 RA therapy on pancreatic health is crucial for clinicians and patients alike, guiding informed decision-making in diabetes management. Therefore, our study aims to assess the risk of pancreatic cancer in patients with T2DM who are being treated with GLP-1 RAs.

## 2. Materials and Methods

### 2.1. Inclusion and Exclusion Criteria

The Institution Board Review Committee at the Charleston Area Medical Center has approved this study (IRB Number: 23-956). Written informed consent from patients was waived due to the deidentified nature of the TriNetX clinical database. The TriNetX (Cambridge, MA, USA) database is a global federal research network that combines real-time data with electronic medical records. This study was conducted using the TriNetX database through the US Collaborative Network, which comprises 63 Healthcare Organizations (HCOs) from the United States of America [3]. Patients with type 2 diabetes mellitus were identified and divided into patients receiving GLP-1 RAs and those who are not. We excluded patients with pancreatic cysts, specific genes associated with pancreatic cancer, which include BRCA1, BRCA2, CDKN2A, KRAS, MEN1, MLH1 MSH2, NOTCH1, PALB2, PMS2, and PRSS1S, a family history of the previously mentioned genes, a family history of pancreatic cancer, and Von Hippel–Lindau syndrome. We also excluded patients above the age of 90. The prior exclusion criteria were identified with specific ICD-10 and genetic codes. A list of all codes used in our study is highlighted in the Appendix A. We compared the rate of pancreatic cancer between the two cohorts over 7 years.

### 2.2. Statistical Analysis

Adult patients aged ≥18 years with type 2 diabetes mellitus in real-time data were identified. Patients with type 2 diabetes mellitus who were included in our study were divided into two cohorts: the first cohort included patients who were being treated with a GLP-1 RA and the second cohort included patients who were not being treated with a GLP-1 RA. Following this, propensity score matching (PSM) of both cohorts to ensure successful and effective balancing was completed. TriNetX conducts logistic regression analysis to obtain propensity scores for each cohort. PSM was performed using patients’ baseline demographics, comorbidities, lab values, and medications received. Demographics included in PSM included age at index, race, gender, and body mass index (BMI). Comorbidities included were tobacco use, history of chronic pancreatitis, prior acute pancreatitis, and alcohol use disorder. Lab values included were hemoglobin A1C levels. Medications used in PSM were metformin, insulin, and dipeptidyl peptidase-4 (DPP-4) inhibitors. A study flow diagram is shown below in Figure 1.

Following propensity score matching, an analysis of the outcomes was conducted. Cumulative Incidence curves and log-rank tests were employed to investigate the correlation of GLP-1 RA use and the risk of pancreatic cancer between groups. Risk ratios (RRs) with their respective 95% confidence intervals (CIs) were calculated for the outcomes. Statistical significance was determined at a *p*-value of less than 0.05. The statistical analyses were conducted using the TriNetX platform.

## 3. Results

### 3.1. Baseline Characteristics

We identified 7,146,016 patients with type 2 diabetes mellitus. Of those, 10.3% (*n* = 736,015) were being treated with a GLP-1 RA. The remaining 89.7% (*n* = 6,410,000) were not being treated with a GLP-1 RA. The mean age in the GLP-1 RA group was 54.4 with a standard deviation (SD) of 12.8 as compared to 60.9 with an SD of 15.3 in the non-GLP-1 RA receivers. More than half the patients receiving GLP-1 RAs were females 52.6% as compared to 47.3% in the other group. In total, 61.7% of the GLP-1 RA receivers were white as compared to 57.8% of the non-receivers. The mean body mass index (BMI) ratio in the GLP-1 RA group was 35.3 with an SD of 6.8 with a mean body weight of 228.5 with an SD of 59.6, as compared to a BMI of 30.8 with an SD of 7.2 and a body weight of 200 with an SD of 55.4 in the group that was not treated with a GLP-1 RA. Of the GLP-1 RA group, 0.5% had a history of acute pancreatitis and 0.1% had chronic pancreatitis as compared to 0.9% and 0.4%, respectively, in the other group. In total, 2.1% of the GLP-1 RA group were smokers and 0.2% used alcohol as compared to 1.9% and 0.5%, respectively. The mean hemoglobin A1C in the GLP-1 RA group was 7.9 with an SD of 2.2 as compared to 7.3 with an SD of 2.1 in the other group. Of the GLP-1 RA group, 39.5% were on metformin, 24.1% were on insulin, and 7% were on dipeptidyl peptidase-4 (DPP-4) inhibitors. On the other hand, of the group that did not receive GLP-1 RAs, 19.8% were on metformin, 24.1% were on insulin, and 3.1% were on DPP-4 inhibitors.

### 3.2. Outcomes

After propensity score matching (PSM), our cohort consisted of 1,442,220 patients. It was divided evenly into two groups: those who received GLP-1 RA and those who did not. There was no statistically significant difference between the two groups in the above-mentioned variables. In the GLP-1 RA group, the mean age was 54.4 with an SD of 12.8. More than half the cohort were females (52.6%) and more than half the cohort was white 61.7%. The mean BMI was 35.3 with an SD of 6.8, and the mean weight was 228.5 with an SD of 59.6. In total, 0.5% had a history of acute pancreatitis and 0.1% had chronic pancreatitis. In the GLP-1 RA group, 2.1% were smokers, and 0.2% used alcohol. The mean hemoglobin A1C level was 7.9 with an SD of 2.2. In terms of medications, 39.5% were on metformin, 24.1% were on insulin, and 7% were on DPP-4 inhibitors. A full list of PSM components before and after matching is shown below in Table 1.

After PSM, we compared the rate of pancreatic cancer between both groups after 7 years of therapy. After 7 years of therapy, the cancer risk between the two groups was noted to be lower in the GLP-1 RA group. In the GLP-1 RA group, it was 0.14% as compared to 0.2% (*p*-value < 0.0001). The risk ratio of developing pancreatic cancer in the GLP-1 RA group was 0.69 with a 95% confidence interval of (0.639, 0.752). Consequently, patients who received GLP-1 RA had a calculated pancreatic cancer risk reduction of 31%, highlighting a possible pancreatic-protective effect of GLP-1 RA. A summary of results, log-rank tests, and hazard ratios is highlighted below in Table 2 with the cumulative event curve in Figure 2 highlighting the difference. 

## 4. Discussion

### 4.1. Panceatic Cancer Incidence and Prevalanece

In the United States of America (U.S.A.), the estimated number of cases of new pancreatic cancer in 2024 was more than 65,000 cases [4]. The incidence continues to rise at a rate of up to 1% per year, and it is expected to be the second leading cause of cancer death in the U.S.A. by 2030 [5,6]. Unfortunately, there is no effective screening method for pancreatic cancer, making its 5-year survival rate in 2020 up to 10%, which is higher than previously reported in 2000 at approximately 5% [6]. Pancreatic cancer is usually diagnosed at the age of 71 in the U.S.A., with a slightly higher incidence in women compared to men [7]. At the time of presentation, half the patients already have metastatic disease, with only 10–15% having surgery-amendable disease [6]. In 2019, the United States Preventive Services Task Force (USPSTF) continued to recommend against screening for pancreatic cancer as there was no evidence that screening improves disease-related morbidity or mortality and that the risks of screening outweigh the benefits [8].

### 4.2. Risk Factors

Risk factors for pancreatic cancer can be divided into modifiable and inherited [9]. Some of the strongest modifiable risk factors include smoking, alcohol use, and chronic pancreatitis. Pathogenic germline gene variants associated with pancreatic cancer were found in up to 9.7% of these cases [10,11,12].

#### 4.2.1. Tobacco Use

Smoking is the most common modifiable risk factor, identified in up to 35% of these patients with pancreatic cancer. The pathogenic mechanism involves mutations in the KRAS and p53 genes. Additionally, smoking causes a state of chronic inflammation. Subsequently, there is a release of cytokines and growth factors that induce cellular proliferation, leading to pancreatic cancer development [13].

#### 4.2.2. Alcohol Use

Some studies showed that the consumption of more than three drinks daily increases the relative risk of developing pancreatic cancer up to 1.36 [14]. Alcohol and its metabolites act as pro-carcinogens via chronic inflammation and gene instability [14]. Furthermore, alcohol consumption leads to the development of chronic pancreatitis, which itself is a risk factor for pancreatic cancer development.

#### 4.2.3. Chronic Pancreatitis

Changes in chronic pancreatitis are similar to those of pancreatic cancer on the morphological, functional, and molecular levels. Cytokines such as Tumor Necrosis Factor (TNF) alpha, interleukin-6, interleukin-8, platelet-derived growth factor (PDGF), and Transforming Growth Factor (TGF) beta induce non-surveilled cellular proliferation [15]. The risk of pancreatic cancer is elevated in chronic pancreatitis irrespective of sex or country with a reported relative risk of 13.3 in a meta-analysis by Raimondi S. et al. [16,17].

#### 4.2.4. Obesity

Another rising risk factor of pancreatic cancer is obesity. A meta-analysis conducted in 2007 showed a 1.12 relative risk increase per 5 kg/m^2^ in the body mass index (BMI) [18]. A temporal relationship between BMI and pancreatic cancer development risk was seen in a study from 2009 [19]. The pathogenic mechanism of obesity is the chronic macrophage production of pro-inflammatory cytokines leading to abnormal cell proliferation [20]. Furthermore, obesity causes hormonal dysregulation manifested in high levels of leptin and low levels of adiponectin, both associated with a higher risk of pancreatic cancer [20]. Additionally, a meta-analysis conducted in 2012 revealed a potential pathogenic effect of a fatty diet in the development of pancreatic cancer [21]. 

#### 4.2.5. Diabetes Mellitus

The majority of patients with pancreatic cancer were found to develop diabetes shortly prior to diagnosis, which would suggest that diabetes is a consequence of cancer [22]. However, a more recent meta-analysis showed a 1.8 odds ratio of pancreatic cancer development in patients with type 2 diabetes [23]. The mechanism of pathogenesis in type 2 diabetes includes hyperinsulinemia along with high levels of Insulin-like Growth Factor (IGF) 1, which lead to pancreatic cell proliferation [24]. Additionally, they lead to a dysregulation of Pancreatic Stellate Cells and Tumor-associated Macrophages, which subsequently leads to the fibrosis, cellular proliferation, apoptosis inhibition, and hyperplasia that is seen in pancreatic cancer [24,25]. 

#### 4.2.6. Hereditary Factors

Hereditary breast and ovarian cancer syndrome is one of the syndromes associated with pancreatic cancer. The BRCA1 and BRCA2 mutations lead to malignant tumors in the breast, ovary, and pancreas. This association was found in up to 19% of hereditary pancreatic cancers [26]. 

Familial Atypical Multiple Mole Melanoma syndrome is another syndrome associated with pancreatic cancer. This syndrome’s pathology is secondary to a CDKN2A gene mutation, which leads to dysregulation of the normal cellular cycle. Patients with this mutation were reported to have an up to 22-fold increase risk of pancreatic cancer [27].

KRAS gene mutations occurred in almost 86% of tumor samples in four large-scale studies [28,29,30,31]. Mutations in KRAS are an early oncogenic event in pancreatic cancer and are thought to be an initiating event [32]. KRAS mutations affect cell cycle survival and metabolism, cytoskeleton and cell motility, and transcriptional programs, all of which play a role in early pancreatic cancer changes and its progression [32].

Multiple Endocrine Neoplasia (MEN) syndrome type 1 is another genetic disorder that affects multiple endocrine glands including the pancreas [33]. This syndrome is caused by a germline mutation in the MEN1 tumor suppressor gene and has a prevalence of 2–3 per 100,000 [34,35]. About 40% of patients with MEN1 develop cancer in the digestive tract [33]. Malignant neuroendocrine tumors caused by MEN1 gene mutations are associated with worse prognosis and are the most common cause of death in patients with MEN1 [36,37,38]. 

Patients with Hereditary Non-Polyposic Colorectal Cancer or Lynch syndrome were also predisposed to pancreatic cancer with a relative risk of 8.6 [39]. This is due to microsatellite instability caused by mutations in the MSH2, MLH1, and PMS2 genes [9]. 

The NOTCH1 gene is another gene that plays an oncogenic role in pancreatic cancer. Some studies showed that it can promote pancreatic intraepithelial neoplasia [40,41]. NOTCH1 stimulates cell invasion and metastasis in pancreatic cancer cells [41]. This was evidenced by multiple studies showing increased NOTCH1 expression in pancreatic cancer cells [42,43,44,45,46,47,48]. 

Mutations in PALB2 genes increase the risk of pancreatic duct adenocarcinoma [49]. A very recent study by Principe D. showed that the composite mutation rate for PALB2 mutation in patients with pancreatic duct carcinoma was 0.54% [50]. While mutations in PALB2 genes are known to be oncogenic, the impact on the hazard ratio for pancreatic cancer remains unknown [50]. 

In 80% of hereditary pancreatitis, a genetic mutation in PRSS1 can usually be identified [9]. Hereditary pancreatitis is characterized by recurrent acute pancreatitis occurring in childhood. The pathogenic mechanism of this condition occurs due to pancreatic chronic inflammation [51,52]. A study in 2010 found an increased relative risk of 69 for pancreatic cancer development in patients with hereditary pancreatitis compared to the general population [17]. 

#### 4.2.7. Pancreatic Cysts

Pancreatic cysts are known precancerous lesions for pancreatic adenocarcinoma [53]. Pancreatic cysts are not always premalignant; however, there is a 5–8% risk of malignant transformation over 5 to 10 years [54]. This risk was found in patients irrespective of their history of acute or chronic pancreatitis. A list of mentioned risk factors for pancreatic cancers is shown below in Figure 3.

### 4.3. GLP-1 RA Mechanism of Action

GLP-1 is an incretin hormone that is inactivated by dipeptidyl peptidase-4 (DPP-4), which stimulates insulin secretion after meal ingestion. It is released from the L-cells of the small intestine and binds to specific GLP-1 receptors that are expressed in various tissues including pancreatic beta cells and pancreatic ducts [55,56]. GLP-1 exerts its main effect by stimulating glucose-dependent insulin release from pancreatic islet cells [55]. Subsequently, it leads to slow gastric emptying, inhibits inappropriate glucagon release, and increases satiety [57,58,59]. 

Patients with type 2 diabetes have an impaired insulin response to GLP-1 that is believed to be secondary to reduced post-prandial GLP-1 secretion [60]. Similar to native GLP-1, synthetic GLP-1 RAs bind to GLP-1 receptors and stimulate glucose-dependent insulin release from the pancreatic islet cells as their main hypoglycemic role. Additionally, synthetic GLP-1 RAs are resistant to degradation by DPP-4; therefore, they have a longer half-life, which plays a part in their ease of clinical administration. Synthetic GLP-1 RAs can be administered once daily or once weekly. A list of all commercially available GLP-1 RAs are shown in Table 3 below.

### 4.4. GLP-1 RA Safety Profile; Benefits and Side Effects

The most common side effects of GLP-1 RAs are usually seen in the digestive tract [61]. The most common symptoms are nausea, diarrhea, vomiting, constipation, abdominal pain, and dyspepsia. Symptoms are typically not seen with early treatment initiation; however, the frequency increases as therapy is continued. Nausea is the most common side effect reported with GLP-1 RAs in up to 50% of patients [61,62]. 

Current available data do not indicate any increase in significant adverse cardiovascular events with the use of GLP-1 RAs [63,64]. A meta-analysis of 22 trials showed a small increase in heart rate with GLP-1 RA use [65]. However, current evidence and guidelines show a cardiovascular benefit from their use.

Since GLP-1 RAs are synthetic peptides that are typically injected subcutaneously, they can lead to antibody formation, which may potentially lead to allergic reactions and hypersensitivity. Antibody formation was noted with different GLP-1 RAs; however, it did not seem to increase immune-related adverse effects or affect their safety profile [66]. 

Injection site reactions such as rash, erythema, or itching are common with GLP-1 RA use. This was reported in clinical trials of exenatide, lixisenatide, and albiglutide [61]. Injection site reactions were reported more frequently with long-acting GLP-1 RAs compared to short-acting ones [67]. These reactions were transient and did not cause discontinuation of treatment [61].

The risk of hypoglycemic events with GLP-1 RA use was mostly noted when combined with sulphonylurea and insulin [61,68]. This effect was not seen when combined with metformin [61].

Upper respiratory tract and urinary tract infections were reported with GLP-1 RA use, with influenza, cystitis, nasopharyngitis, and viral infections being most common in their individual trials. However, no cause–effect association was found [61].

Headaches were also often reported in the trials; however, they did not lead to therapy discontinuation [61].

Acute kidney injury was reported in GLP-1 RA trials and was believed to be secondary to volume contraction due to GI symptoms or concomitant use of drugs working on the renin–angiotensin pathway [69]. Therefore, we should be cautious when signs of volume depletion develop. However, more recent evidence shows renoprotective evidence and slowing of chronic kidney disease progression as seen in a meta-analysis released in 2023 [70]. They also reduce the progression of albuminuria [71].

Data on the cardiovascular benefits of GLP-1 RAs are widely available [72]. This translated into the strong recommendation of the European Society of Cardiology on the use of GLP-1 RAs in patients with T2DM and atherosclerotic cardiovascular disease (ASCVD) in 2019 [73]. Furthermore, in 2020, the American College of Cardiology (ACC) recommended their use in patients with T2DM who are at risk for ASCVD, heart failure, or kidney disease [74]. More recently, the American Diabetes Association (ADA), in their 2022 guidelines, added GLP-1 RA to the standard medical care of diabetes and recommended their use in T2DM with ASCVD or with a high risk of ASCVD [75].

### 4.5. Association of GLP-1 RAs with Different Cancers

The association of GLP-1 RAs with thyroid cancer is a growing topic. Studies on rodents showed that the stimulation of GLP-1 receptors on thyroid parafolicular C cells stimulates cAMP-dependent synthesis and secretion of calcitonin and, upon longer-term exposure, causes the proliferation of C cells and the formation of C-cell adenomas and carcinomas [76,77]. Previous data showed insufficient evidence that GLP-1 RAs cause the same in humans, possibly due to the lower distribution of GLP-1 receptors on thyroid cells in humans. However, Bezin J. et al. recently published a study on 2562 case subjects that showed an increased risk of all thyroid cancers (adjusted HR 1.58, 95% CI 1.27–1.95) and medullary thyroid cancer (adjusted HR 1.78, 95% CI 1.04–3.5) after 1–3 years of GLP-1 RA use [78]. When analyzing these data, the fact that type 2 diabetes and obesity by themselves are risk factors for thyroid cancer, and that frequent ultrasound monitoring before and after prescribing this class of medications can affect the numbers, should be considered. Many authors agree that the potential risks and benefits should be weighed before prescribing these medications, and further studies are needed.

The association of obesity with cancers and the known effects of GLP-1 RAs on weight loss and immune functions triggered multiple research projects on the positive effects of these medications related to cancer diagnosis. One new retrospective study showed that GLP-1 RAs were associated with a reduced risk of colorectal cancer with and without obesity, suggesting potential protective effects mediated by weight loss and other effects not related to weight loss [79]. Some preclinical studies showed that GLP-1 RAs can inhibit proliferation, increase apoptosis of prostate cancer cells, and decrease inflammation in prostate cancer [80]. A similar effect was seen in NASH and hepatocellular cancer. NASH is a progressive form of NAFLD that can lead to cirrhosis and hepatocellular carcinoma (HCC) formation, and it is rapidly becoming a leading cause of end-stage liver disease and liver transplantation. Studies have shown that GLP-1 RAs not only reduce subcutaneous and visceral adipose tissue but intrahepatic adipose tissue as well. They also decrease hepatic inflammation and injury in advanced NASH. In terms of HCC, they have shown potential benefit by inhibiting tumor cell growth, potentiating apoptosis through the cAMP-PKA-EGFR-STAT 3 axis. Similarly, by affecting different molecular pathways, they can play a role in other cancers, such as endometrial and breast cancers [81].

### 4.6. Association of GLP-1 RA with Pancreatic Cancer and Theories

Concerns for the pancreatic cancer incidence in patients receiving GLP-1 RAs were raised following several publications in 2011 and 2013 [2,82]. This led the Food and Drug Administration (FDA) to issue a warning on pancreatic safety in 2016 [1]. This remained their stance in their most recent official releases in 2019 and 2024 under the warning section [83,84].

The pathophysiology of pancreatic cancer development with GLP-1 RA use is thought to be secondary to chronic low-grade inflammation and proliferative changes leading to KRAS and p53 gene mutations [85]. Without treatment with GLP-1 RAs, patients with T2DM have an increased risk for both pancreatitis and pancreatic cancer. Any mutation, in addition to pre-existing chronic inflammation caused by long-standing T2DM, can tilt the balance towards the progression of neoplasia, which was seen in early animal studies [85]. In addition to inflammation, GLP-1 RAs induce the proliferation of pancreatic islet cells. Acinar cells and ductal cells proliferate in response to GLP-1 RAs [86]. This proliferation may be sufficient to initiate carcinogenesis and eventual pancreatic cancer [85].

### 4.7. Our Study and Current Evidence

Current literature evidence about GLP-1 RAs’ effect on pancreatic cancer remains controversial. With our study findings, we aim to add to the pre-existing data supporting GLP-1 RA use and revoking the association of pancreatic cancer with their use. In fact, our study found a possible protective effect of GLP-1 RAs against pancreatic cancer with a 30% risk reduction over the course of 7 years.

Dankner et al. showed in a recent retrospective study that GLP-1 RA use was not associated with an increased risk of pancreatic cancer [87]. Their findings were very similar to our study. They were unable to find an increased risk of pancreatic cancer over 7 years following the start of GLP-1 RAs. Furthermore, a recent meta-analysis published in Italy in July 2023 did not show an association of pancreatic cancer with GLP-1 RA use [88]. Our study adds to those recent studies’ additional evidence of GLP-1 RA safety.

### 4.8. Study Strengths and Limitations

Our study has several strengths. Firstly, the size of our patient cohort from a nationwide multi-institutional database increases the statistical power of our analysis and allows the generalizability of our findings. Secondly, the selective nature of our exclusion criteria minimizes the confounding bias of any potential risk factor for pancreatic cancer. Additionally, the use of PSM mitigates additional selection bias, ensuring the comparability of both cohorts. Third, the ability to reproduce findings consistent with recently published studies confirms the accuracy of the findings.

Our study does not come without limitations. Firstly, our database is U.S-based, which may affect interpretation and applicability outside of the U.S.A. as it does not take international demographics into account. Secondly, due to the nature of the database and inherent weakness within the electronic health record study design, we were not able to account for the duration of therapy or compliance with treatment. Therefore, future prospective studies are needed to validate our findings.

## 5. Conclusions

The use of GLP-1 RAs in patients with type 2 diabetes mellitus (T2DM) does not increase the risk of pancreatic cancer; in fact, it appears to exert a protective effect over the course of 7 years. This finding supports the continued use of GLP-1 RAs as a therapeutic option in managing T2DM as opposed to the current FDA warning. Our study suggests a pancreatic-protective effect of GLP-1 RAs in addition to their cardioprotective and renoprotective properties. This should further support their use to decrease long-term complications in patients with T2DM.

## Figures and Tables

**Figure 1 cancers-16-01625-f001:**
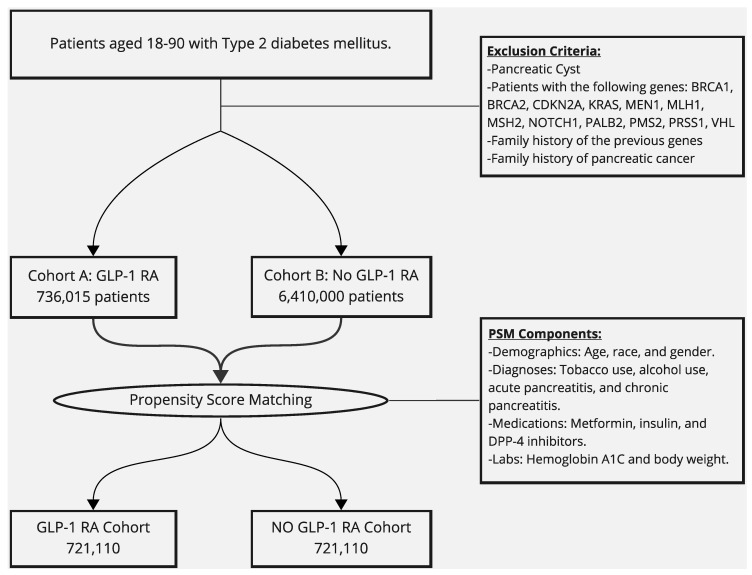
Study design flow graph showing our inclusion criteria and number of patients, exclusion criteria, propensity score-matching (PSM) components, and final count of both cohorts’ patients after PSM.

**Figure 2 cancers-16-01625-f002:**
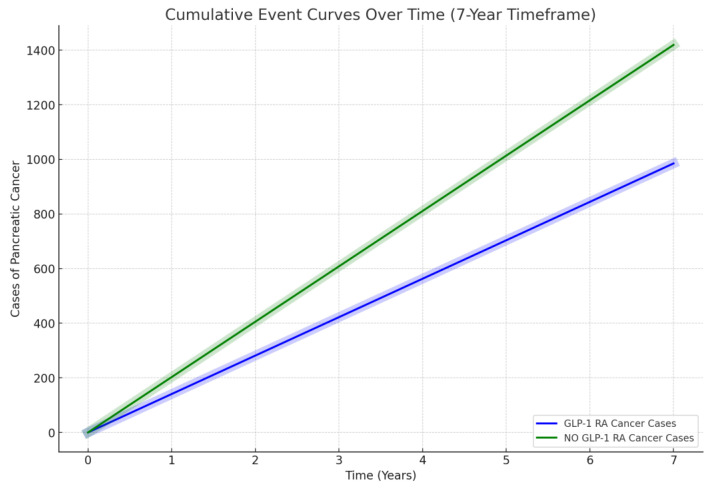
Linear comparison between cumulative pancreatic cancer risk events over 7 years between patients with T2DM on GLP-1 RAs (blue) and patients with T2DM not on a GLP-1 RA (green).

**Figure 3 cancers-16-01625-f003:**
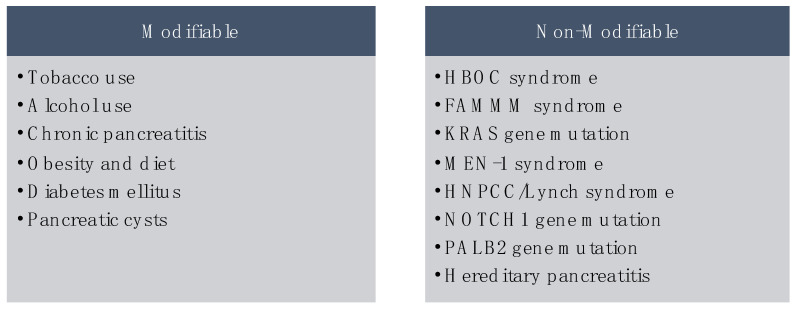
Modifiable and non-modifiable risk factors for pancreatic cancer.

**Table 1 cancers-16-01625-t001:** Patients’ demographics, diagnoses, medications, and lab values before and after PSM.

	Before PSM			After PSM		
Characteristic	GLP-1 RA	No GLP-1 RA	*p*-Value	GLP-1 RA	No GLP-1 RA	*p*-Value
Demographics						
Age	54.4 ± 12.8	60.9 ± 15.3	<0.001	54.4 ± 12.8	54.4 ± 12.9	0.46
White	61.7%	57.8%	<0.001	61.7%	61.7%	0.44
Black or African American	17.9%	16.5%	<0.001	17.9%	18%	0.28
Asian	3%	4.4%	<0.001	3%	3.3%	0.41
Unknown race	12.3%	16%	<0.001	12.3%	12.3%	0.88
Female	52.6%	47.3%	<0.001	52.6%	52.7%	0.75
Diagnosis						
Tobacco use	2.1%	1.9%	<0.001	2.1%	2%	0.12
Alcohol use	0.2%	0.5%	<0.001	0.2%	0.2%	0.09
Acute pancreatitis	0.5%	0.9%	<0.001	0.5%	0.5%	0.06
Chronic pancreatitis	0.1%	0.4%	<0.001	0.1	0.1	0.08
Medication						
Metformin	39.5%	19.8%	<0.001	39.5%	39.5%	0.85
Insulin	24.1%	20.3%	<0.001	24.1%	24%	0.39
DPP-4 inhibitors	7%	3.1%	<0.001	7%	6.9%	0.32
Laboratory Results						
Hemoglobin A1c	7.9 ± 2.2	7.3 ± 2.1	<0.001	7.9 ± 2.2	7.5 ± 2.2	0.062
Body weight	228.5 ± 59.6	200 ± 55.4	<0.001	228.1 ± 59.6	210 ± 58.2	0.078

**Table 2 cancers-16-01625-t002:** Summary of results, log-rank test, and hazard ratio.

	GLP-1 RA (*n* = 720,678)		NO GLP-1 RA (*n* = 720,678)		* p * -Value
Outcome Pancreatic cancer	0.14% (*n* = 985)		0.2% (*n* = 1419)		<0.0001
Log-rank test	X^2^		df		*p*
	250.681		1		0.000
Hazard ratio and proportionality	HR	95% CI	X^2^	df	*p*
	0.524	(0.483–0.568)	41.010	1	0.000

**Table 3 cancers-16-01625-t003:** List of commercially available GLP-1 RAs.

Generic Name	Brand Name(s)	Dosing Schedule
Exenatide	Byetta, Bydureon	Byetta: twice daily; Bydureon: once weekly
Liraglutide	Victoza, Saxenda	Once daily
Dulaglutide	Trulicity	Once weekly
Semaglutide	Ozempic, Rybelsus, Wegovy	Ozempic: once weekly; Rybelsus: once daily; Wegovy: once weekly
Albiglutide	Tanzeum	Once weekly
Lixisenatide	Adlyxin, Lyxumia	Once daily

## Data Availability

Available data are presented within the paper. Additional data are only available as permitted by third parties.

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
