# Peer review of "The Use of Glucagon-like Peptide-1 Receptor Agonists in Patients with Type 2 Diabetes Mellitus Does Not Increase the Risk of Pancreatic Cancer: A U.S.-Based Cohort Study"

_cancers, 2024, doi:10.3390/cancers16091625_

Round 1

Reviewer 1 Report

Comments and Suggestions for Authors

This study investigated the association between a type 2 diabetes medication (GLP-1 RA) and pancreatic cancer risk. Researchers analyzed data from a large database encompassing millions of patients with type 2 diabetes in the United States. They divided the patients into two groups: those receiving GLP-1 RA medication and those who weren't.  After carefully excluding patients with known pancreatic cancer risk factors, the researchers used a statistical technique to create comparable groups between those on and off the medication. Finally, they compared the rates of pancreatic cancer between the two groups over a seven-year follow-up period.

The findings suggest that GLP-1 RA use in patients with type 2 diabetes is not linked to an increased risk of pancreatic cancer. In fact, the study indicates a potentially reduced risk for those taking the medication compared to those who don't. These results are encouraging for the safety profile of GLP-1 RA medications in type 2 diabetes management.

Please address the following minor concerns:

Please include the rationale behind this study and what you aim to investigate at the end of introduction section.

1.        Minor English editing is required. For ex.:

a.       Line 20: Change "are" to "is" because "GLP-1 RA" is singular.

b.       Line 307: Spell correction “these”.

3.      Methods:

a.      Please rearrange the sections 2.2 should come before, then 2.1 Statistical Analysis. Also, mentions the groups which were studied in the section: Inclusion/exclusion not in the stats analysis. Which method for PSM was followed: logistic regression or Classification and Regression Tree Analysis.? Please elaborate the statistical analysis

b.     In inclusion criteria, please explain why only three glucose lowering medicines were considered. What about α-glucosidase inhibitors, rosiglitazone, sulfonylureas, and meglitinides?  Were all the patients included in the study on three mentioned GLMs?

c.      Was ethnicity also taken into consideration? Was there non-significant difference between different age groups?

4.      Please rearrange the table 2 from the graph of cumulative events over time and add a figure legend. It is hard to interpret this table with graph.

5.      Add a figure legend with Figure 1. The study flow graph is unclear-please including the details like PSM, inclusion, explaining each arm of the flow chart.

The results discussed lack

6.      Line 370: Author mentions current FDA warning but that was back dated to 2016. Infact, FDA has recently approved the use of GLP-1 RA semaglutide for Type 2 Diabetes or Weight Loss.

7. Please elaborate the conclusion section with strong argument.

Comments on the Quality of English Language

Minor English editing is required.

Author Response

Thank you so much for reviewing our study! We truly appreciate your timeliness, time, and effort!

"Please include the rationale behind this study and what you aim to investigate at the end of introduction section"

Thank you for highlighting this. We implemented your suggestion.

1. "Minor English editing is required. For ex.:

      • Line 20: Change "are" to "is" because "GLP-1 RA" is singular.
      • Line 307: Spell correction “these”."

Thank you for highlighting these typing mistakes. We made the necessary changes and proofread the manuscript and fixed some other ones as well.

2. Methods:

    • "Please rearrange the sections 2.2 should come before, then 2.1 Statistical Analysis. Also, mentions the groups which were studied in the section: Inclusion/exclusion not in the stats analysis. Which method for PSM was followed: logistic regression or Classification and Regression Tree Analysis.? Please elaborate the statistical analysis"

Thank you for the feedback! We re-ordered section 2 as suggested. TriNetX uses logistic regression analysis for PSM. We went ahead and elaborated in the methods section. 

    • In inclusion criteria, please explain why only three glucose lowering medicines were considered. What about α-glucosidase inhibitors, rosiglitazone, sulfonylureas, and meglitinides?  Were all the patients included in the study on three mentioned GLMs?

We did not only include those 3 GLMs. We included any patient with T2DM with first cohort being on GLP-1 RA and second cohort who is not on GLP-1 RA. The abovementioned 3 GLMs are used in PSM, but no therapy was excluded. Those 3 GLMs are the ones most commonly associated with pancreatitis/pancreatic cancer development and hence they are used in PSM to avoid confounders. We adjusted our study flow graph to highlight our exclusion criteria and PSM components.

    • Was ethnicity also taken into consideration? Was there non-significant difference between different age groups?

Ethnicity and race were included in PSM. We went ahead and added those to Table 1. Due to the nature of TriNetX database, we are unable to further stratify the age into categories, however, after PSM there was no statistically significant difference between the two cohorts as shown in Table 1, ensuring comparable cohorts.

3. "Please rearrange the table 2 from the graph of cumulative events over time and add a figure legend. It is hard to interpret this table with graph."

Thank you for pointing that out! We went ahead and made Table 2 separate from Figure 2 as suggested to avoid confusion. We also added descriptive legend to the figure.

4. "Add a figure legend with Figure 1. The study flow graph is unclear-please including the details like PSM, inclusion, explaining each arm of the flow chart."

Thank you for pointing that out. We went ahead and re-designed the graph by adding PSM and its components as well as descriptive legend.

5. "Line 370: Author mentions current FDA warning but that was back dated to 2016. Infact, FDA has recently approved the use of GLP-1 RA semaglutide for Type 2 Diabetes or Weight Loss."

Excellent remark and it is correct. The FDA approved it for T2DM or weight loss, however, the pancreatic warning stands even in their most recent press release in March of 2024 about Wegovy (semaglutide) and 2019 about Rybelsus (semaglutide). The official release of 2016 specifically addresses pancreatic involvement which is the main topic of our study. We also cited the 2019 and 2024 official FDA statements under section 4.6 to highlight that.

6. "Please elaborate the conclusion section with strong argument."

Thank you for the feedback. We expanded our conclusion section.

Once again, we would like to thank you for your time and effort! Your feedback allows us to improve our manuscript and ensures the accuracy and quality of our manuscript! Thank you.

Reviewer 2 Report

Comments and Suggestions for Authors

The authors present a retrospective national cohort study aimed at assessing the risk of developing pancreatic cancer in patients with type 2 diabetes mellitus (T2DM) who are undergoing treatment with a Glucagon-like Peptide-1 receptor agonist. Among the cohort, 10.3% (n=736,015) were receiving treatment with a GLP-1 RA, while the remaining 89.7% (n=6,410,000) were not receiving such treatment, based on data extracted from the TriNetX clinical database (Cambridge, MA, USA).

I would like to raise the following concerns.

1.

Lines 74-75 (page 3) indicate that “Cumulative Incidence curves and log-rank tests were employed to investigate disparities in all-cause mortality rates between groups”. However, it seems that the results of this analysis are not presented in the text.

Alternatively, the log-rank tests may have been used to assess the correlation analysis of the use of Glucagon-Like Peptide-1 Receptor and risk of pancreatic cancer.

2.

The authors mentioned "Tobacco Use, Alcohol Use, Chronic Pancreatitis, Obesity and Diet, Diabetes Mellitus, Pancreatic Cysts" as modifiable factors. However, Table 1 does not include information about these modifying factors except for Tobacco Use and Chronic Pancreatitis.

3.

In Table 1, categorical variables such as Tobacco Use, Acute Pancreatitis, Chronic Pancreatitis, Metformin, Insulin, and DPP-4 Inhibitors, as well as continuous variables like Hemoglobin A1c and Body Weight, if categorized, would suggest conducting a subgroup analysis.

Author Response

Thank you for reviewing our study! We appreciate your comments and timeliness!

1. "Lines 74-75 (page 3) indicate that “Cumulative Incidence curves and log-rank tests were employed to investigate disparities in all-cause mortality rates between groups”. However, it seems that the results of this analysis are not presented in the text. Alternatively, the log-rank tests may have been used to assess the correlation analysis of the use of Glucagon-Like Peptide-1 Receptor and risk of pancreatic cancer."

Thank you so much for pointing that out! We made the necessary changes to the phrasing as suggested under section 2.2. We also re-designed our outcomes table "Table 2" to include the used Log-Rank test and HR. 

2. "The authors mentioned "Tobacco Use, Alcohol Use, Chronic Pancreatitis, Obesity and Diet, Diabetes Mellitus, Pancreatic Cysts" as modifiable factors. However, Table 1 does not include information about these modifying factors except for Tobacco Use and Chronic Pancreatitis."

Thank you again for pointing that out. We added alcohol use to the table. Body weight and Hemoglobin A1C levels are both included as well. Pancreatic cysts are part of the exclusion criteria, therefore, we did not add them to PSM components. Table 1 should be updated now.

3. "In Table 1, categorical variables such as Tobacco Use, Acute Pancreatitis, Chronic Pancreatitis, Metformin, Insulin, and DPP-4 Inhibitors, as well as continuous variables like Hemoglobin A1c and Body Weight, if categorized, would suggest conducting a subgroup analysis."

That is an excellent point and is of interest to us! However, the nature of our TriNetX database is code-based and it does not allow for subgroup analysis without altering our study structure i.e., our inclusion, exclusion, and PSM components. That would be a great idea for our future study to use each co-variate separately.

Once again, we would like to thank you for your time and effort. Your feedback helps improve the quality of our manuscript. 

Thank you.

Round 2

Reviewer 2 Report

Comments and Suggestions for Authors

All the concerns have been answered.